# Hematopoietic Dysfunction during Graft-Versus-Host Disease: A Self-Destructive Process?

**DOI:** 10.3390/cells10082051

**Published:** 2021-08-10

**Authors:** Konradin F. Müskens, Caroline A. Lindemans, Mirjam E. Belderbos

**Affiliations:** 1Princess Máxima Center for Pediatric Oncology, 3584 CS Utrecht, The Netherlands; k.f.muskens-2@prinsesmaximacentrum.nl (K.F.M.); c.a.lindemans@prinsesmaximacentrum.nl (C.A.L.); 2Wilhelmina Children’s Hospital, University Medical Center Utrecht, 3584 EA Utrecht, The Netherlands

**Keywords:** hematopoietic stem cell transplantation, graft-versus-host disease, hematopoiesis, bone marrow niche, cytopenia, poor graft function, graft failure

## Abstract

Graft-versus-host disease (GvHD) is a major complication of allogeneic hematopoietic (stem) cell transplantation (HCT). Clinically, GvHD is associated with severe and long-lasting hematopoietic dysfunction, which may contribute to the high mortality of GvHD after HCT. During GvHD, excessive immune activation damages both hematopoietic stem and progenitor cells and their surrounding bone marrow niche, leading to a reduction in cell number and functionality of both compartments. Hematopoietic dysfunction can be further aggravated by the occurrence—and treatment—of HCT-associated complications. These include immune suppressive therapy, coinciding infections and their treatment, and changes in the microbiome. In this review, we provide a structured overview of GvHD-mediated hematopoietic dysfunction, including the targets in the bone marrow, the mechanisms of action and the effect of GvHD-related complications and their treatment. This information may aid in the identification of treatment options to improve hematopoietic function in patients, during and after GvHD.

## 1. Introduction

Allogeneic hematopoietic (stem) cell transplantation (HCT) is a curative therapy for a variety of diseases, including certain immune deficiencies, inborn errors of metabolism and hematologic malignancies [1]. Prior to transplantation, the recipient receives a conditioning regimen consisting of chemo-and/or radiotherapy to ablate his/her own hematopoietic system and to generate space for the donor hematopoietic stem and progenitor cells (HSPCs). In addition, lymphodepleting drugs are given to prevent rejection of donor cells by the host immune system. To facilitate tolerance of the donor-derived immune system to the host, the recipient is treated with immunosuppressive medication, which is slowly tapered over several weeks to months after transplantation.

After HCT, reconstitution of the hematopoietic and immune system is a complex and delicate process, which may take several months up to years to complete [2]. After conditioning, HCT recipients undergo an aplastic period, characterized by severe neutropenia, anemia and thrombocytopenia, resulting in high risk of infections and bleeding complications. Neutrophils are the first blood cells to recover, at a median time of 14–30 days after HCT, depending on the graft source, administered cell dose and use of granulocyte-colony stimulating factor (G-CSF) [3]. Subsequently, T cell recovery ensues 3–6 months after HCT, which relies on peripheral expansion of memory T cells infused with the graft, followed by the de novo production of naive T cells in the thymus. The B cell compartment is the slowest to recover and may take up to several years [3]. The timely reconstitution of donor-derived blood and immune cells is of utmost importance to prevent HCT-related complications and is one of the major predictors of HCT outcome [4].

Even though transplantation protocols have been highly optimized and may be fine-tuned for each individual, HCT remains a high-risk therapy with potentially life-threatening complications and over 30% mortality [5]. One of the major complications of HCT is graft-versus-host disease (GvHD). During GvHD, alloreactive donor T cells recognize the host tissues as non-self, resulting in widespread immune activation and tissue damage (Figure 1) [6]. The clinical severity of GvHD varies from mild skin rash or diarrhea to generalized erythema, extensive intestinal fluid loss and liver dysfunction, which may become fatal. In fact, GvHD is the main cause of transplant-related mortality [6].

Historically, GvHD has been classified as acute or chronic, based on the presence of symptoms within or after the first 100 days after HCT [7]. Acute GvHD (aGvHD) is caused by an excessive immune response of allogeneic T cells to host antigens. Cell-based cytotoxicity and high expression of cytokines such as tumor necrosis factor alpha (TNFα), interferon gamma (IFN-γ) and interleukin 2 (IL-2) result in tissue damage and, eventually, organ dysfunction [8]. The major target organs of aGvHD are the skin, intestine, liver and lung [7]. These preferential targets may be explained by their sensitivity to conditioning-induced tissue damage [9]. Chronic GvHD (cGvHD) represents a persistent inflammatory state, characterized by dysregulated T and B cell immunity and tissue fibrosis, and is often associated with auto-antibody formation [10]. cGvHD is less organ-restricted and may present with a large variety of symptoms, including scleroderma, Sjögren’s syndrome and dysfunction of the gastro-intestinal and musculoskeletal system [11]. This review will focus predominantly on aGvHD.

Over the last few decades, the bone marrow (BM) has been increasingly recognized as an important target of GvHD [12,13]. Consequently, GvHD may affect both hematopoietic reconstitution, as well as hematopoietic function. In retrospective studies, GvHD has been associated with primary failure of donor HSPCs to establish functional hematopoiesis in the recipient [2,14], as well as secondary deterioration of graft function after initial engraftment [14,15,16]. Clinically, GvHD is often accompanied by a decrease in peripheral blood counts, which is also observed in murine studies [17]. GvHD-related hematopoietic dysfunction commonly affects all blood cell lineages, but thrombopoiesis and B cell lymphopoiesis appear most dysregulated [2,16,18,19]. T cell lymphopoiesis can also be affected, but this is likely secondary to GvHD-mediated damage to the thymus and lymphoid organs, rather than the bone marrow [20]. Notably, thymic injury may impair the process of negative selection during T cell development. This enables the development of autoreactive T cells, which could play a role in the autoimmune symptoms observed in cGvHD [13]. The presence of hematopoietic dysfunction during GvHD is an independent predictor of poor outcome of GvHD [2,21]. These findings underline the clinical importance of GvHD-related hematopoietic dysfunction after HCT.

Despite the clinical association between GvHD and hematopoietic dysfunction, the exact mechanisms that lead to hematopoietic dysfunction during and after GvHD remain incompletely understood. In this review, we provide a structured overview of GvHD-related hematopoietic dysfunction. First, we describe the potential targets of GvHD-mediated damage to the bone marrow, including HSPCs and their complex microenvironment, called the bone marrow niche [22]. Next, we look at the mechanisms by which GvHD may damage bone marrow constituents. Finally, we discuss how clinical factors that accompany GvHD, such as prolonged immunosuppressive therapy, infectious complications, and alterations to the microbiome [23], may aggravate hematopoietic dysfunction during GvHD. By highlighting amenable pathways and potential therapeutic targets, this review may help identify treatment options to improve hematopoietic function in patients, during and after GvHD.

## 2. Bone Marrow Targets of GvHD: Hematopoietic Stem Cells and Their Niche

Hematopoietic stem cells (HSC) are the foundation of the hematopoietic system, supplying the progenitor cells that differentiate into all cell types in the blood. During steady state hematopoiesis, HSC slowly proliferate, producing more committed progenitors while maintaining the stem cell pool through self-renewal divisions [24]. HSPCs are tightly regulated within their specialized microenvironment in the bone marrow. This bone marrow niche provides a decor with cell–cell interactions, secreted factors and physical cues that regulate HSPC migration, proliferation, and differentiation [25]. Consequently, GvHD-related hematopoietic dysfunction may result from direct damage to HSPCs, damage to the surrounding bone marrow niche, or both.

### 2.1. GvHD Reduces HSPC Number and Function

Different murine models have been used to investigate the impact of GvHD on HSPCs. Early work mostly made use of donor lymphocyte infusion (DLI) models. In these models, the crossbreed of two different parental mice strains is injected with (often spleen-derived) lymphocytes from either parent. Due to the mismatch in major histocompatibility complex (MHC) molecules, injected lymphocytes recognize recipient cells as non-self. This results in a GvHD-phenotype, characterized by skin and fur changes, weight loss and diarrhea [26,27]. DLI-induced GvHD mice also suffer from hematopoietic dysfunction, including reduced peripheral blood counts with hypocellular bone marrow [26,28]. HSPCs from these mice are reduced in number and exhibit impaired self-renewal capacity in ex vivo colony-forming assays [26,29]. Interestingly, when low levels of lymphocyte are infused, hematopoietic dysfunction persists, while other GvHD symptoms are abated [26,28,30]. These data suggest that the hematopoietic system may be more sensitive to immune-mediated damage compared to other GvHD targets.

In contrast to these DLI models, in the allogeneic HCT setting, both the T cells and the HSPCs are donor derived. Therefore, one might expect these syngeneic HSPCs to be protected from direct GvHD-mediated damage after transplantation. In murine transplantation models, while GvHD is generally absent when T cells are depleted from the transplant, addition of T cells results in a severe GvHD phenotype that quickly becomes fatal [31]. Surprisingly, several studies have shown that GvHD results in decreased HSPC number and function in these models, suggesting that HSPCs are damaged by GvHD in the transplantation setting [32,33,34]. Similarly, bone marrow aspirates from human patients with GvHD show reduced numbers of progenitor cells with impaired proliferation in ex vivo colony forming assays, compared to bone marrow aspirates from patients without GvHD [35]. Together, these data demonstrate that donor HSPCs are affected by GvHD.

One unresolved issue, however, is whether donor HSPCs are damaged by T cells directly. Alternatively, the reduction in HSPC number and functionality may be secondary to GvHD-induced niche dysfunction. Evidence in support of this latter hypothesis comes from a study in which HSPCs of GvHD-suffering mice were retransplanted into a secondary recipient mice. After an initial delay in reconstitution, no difference in hematopoietic function was observed 5 weeks post-HCT compared to mice transplanted with healthy HSPCs [34]. These findings suggest that impaired HSPC function during GvHD is at least partially caused by a damaged or inflamed microenvironment. Importantly, HSPC function may be restored in the absence of these inflammatory surroundings.

### 2.2. GvHD Damages BM Niche Components, Impairing Overall Niche Function

After HCT, the supporting bone marrow niche remains largely of host origin [36] and may be targeted by alloreactive donor T cells. Indeed, bone marrow of mice suffering from GvHD shows extensive infiltration of alloreactive T cells, and reduced number of niche cells, including endothelial cells and osteoblasts [34,37,38]. Similar results are seen in bone marrow biopsies from aGvHD patients, which show fewer mesenchymal cells and osteoblasts compared to transplant recipients without GvHD [18,39].

GvHD not only reduces the cellularity of the niche, but also impairs its function [34,37]. For instance, increased permeability of bone marrow endothelial cells has been observed during murine GvHD [34], which may exacerbate niche damage by facilitating the entry of alloreactive T cells into the bone marrow stroma. Interestingly, murine transplantation of healthy HSPCs into GvHD-damaged niches results in impaired hematopoietic reconstitution of the B lymphoid, and to a lesser extent myeloid lineage, indicating that a damaged niche may be unable to support hematopoiesis [34,37]. Indeed, in patients with cGvHD, reduced osteoblast numbers in the bone marrow niche were found to be associated with otherwise unexplained cytopenias [40]. In summary, GvHD can be associated with reduced cellularity and impaired function of the BM niche, which may contribute to hematopoietic dysfunction.

## 3. Mechanisms of GvHD-Related Hematopoietic Dysfunction

GvHD is characterized by excessive T cell activation and dysregulated cytokine production [7]. Alloreactive T cells exert their functions via cell-cell contact dependent mechanisms and via the production of soluble factors. Below, we will detail how these mechanisms affect hematopoietic function.

### 3.1. Fas-Mediated Cytotoxicity Is Important for GvHD-Mediated Hematopoietic Dysfunction

In general, activated T cells mediate target cell destruction via three mechanisms: perforin/granzyme-mediated cytotoxicity, Fas/Fas-ligand (FasL)-induced apoptosis, or cytokine-mediated cytotoxicity [41] (Figure 2a). During GvHD, all three mechanisms play distinct roles in the destruction of target cells [42]. In mice, T cells deficient in Fas-mediated cytotoxicity induce less hepatic and cutaneous GvHD [43]. By contrast, knockout of the perforin-granzyme pathway results in delayed GvHD onset [43]. T cells deficient in both Fas-mediated and perforin-mediated cytotoxicity still induce severe GvHD, highlighting the importance of Fas-and perforin-independent mechanisms, such as cytokine-mediated cytotoxicity, in GvHD [44,45].

In the bone marrow, the Fas/FasL pathway appears particularly important for T-cell mediated tissue damage [34,37,46]. Fas is a death-receptor of the TNF-receptor family. During Fas-mediated killing, binding of FasL, expressed by T cells, to the Fas receptor on target cells initiates apoptosis in the target cell [41]. In mice, infusion of alloreactive T cells results in bone marrow hypocellularity and hematopoietic dysfunction, which can be prevented by knockout of FasL [34,37,46]. This destruction targets different niche components, including osteoblasts and endothelial cells [34,37]. The perforin-granzyme cytolytic pathway seems to be less involved during T cell-mediated bone marrow damage: infusion of T cells with a perforin knockout results in similar levels of bone marrow hypocellularity and pancytopenia compared to wildtype T cells [46,47].

As previously mentioned, it is still unknown whether GvHD targets donor HSPCs directly. In principle, HSPCs could be subject to Fas-mediated cytotoxicity. Although HSPCs do not express Fas during steady state hematopoiesis, Fas can be upregulated on HSPCs during inflammation [48,49]. Fas-mediated destruction of host HSPCs has been demonstrated in murine DLI and transplantation models [26,50]. In these models, knockout of FasL in T cells prevents the destruction of host HSPCs. These studies show that, despite their immune-privileged environment within the bone marrow niche [51], host HSPCs can be targeted by T cell-mediated destruction, and this destruction is dependent on the Fas/FasL pathway.

Importantly, during GvHD, Fas-mediated cytotoxicity may not require recognition of an MHC-peptide complex on target cells. In mice, knockout studies in which MHC was expressed solely in APCs revealed that MHC is required for initial T cell activation by APCs, but not for subsequent damage to target cells in liver or intestine [52]. Similarly, MHC-independent T cell destruction could target syngeneic donor HSPCs. Unfortunately, research investigating this hypothesis is very limited. One study found that in mice, transplantation of mutant HSPCs, immune to Fas-induced apoptosis, together with wild-type T cells, did not improve GvHD-mediated hematopoietic dysfunction [34]. If direct destruction of donor HSPCs is present during GvHD, this finding suggests that other mechanisms than Fas-mediated cytotoxicity are likely to be involved.

### 3.2. Cytokines Induce Cellular Destruction and Dysfunction of Niche and HSPCs

Cytokines are key players in the pathophysiology of GvHD. Inflammatory cytokines, such as IL-1, IL-6 and TNFα, are produced by APCs and damaged tissue in response to the HCT-conditioning regimen [8]. Subsequently, activated T cells amplify the immune response by secreting additional cytokines, including IFN-γ and IL-2, that direct T cell development and are involved in the cellular destruction and functional impairment in the target organs of GvHD [8]. Similarly, cytokines play several important roles in GvHD-mediated hematopoietic dysfunction, by enhancing cytotoxicity and altering the function of both HSPCs and the bone marrow niche (Figure 2c) [53,54,55,56]. A vast array of cytokines is involved in GvHD-mediated hematopoietic dysfunction, and the exact function of each individual cytokine is still the subject of investigation [57]. Here, the different mechanisms by which cytokines can affect hematopoiesis are discussed conceptually, using some key cytokines as example.

Firstly, cytokines can contribute to the direct destruction of host cells during GvHD. For instance, TNFα and IFN-γ have been shown to directly activate apoptotic pathways in several cell types that are targeted by GvHD [58,59,60]. Indeed, multiple niche components, such as endothelial cells and osteoblasts, also undergo apoptosis in response to high levels of these cytokines [61,62]. TNFα and IFN-γ also induce the expression of Fas and MHC on niche cells, making them more susceptible to Fas-mediated cytotoxicity [53,54]. Although heavily debated, HSPCs may also be susceptible to direct cytokine-induced apoptosis by TNFα and IFN-γ [63,64]. Since HSPCs seem largely unaffected by Fas-mediated destruction, the potential destruction of HSPCs during GvHD could be induced by cytokine-mediated damage instead.

Secondly, cytokines may affect HSPC function by regulating the cell division and self-renewal capacity of HSPCs [65,66,67]. During infection, high cytokine levels result in increased cell-cycle entry of HSCs and a bias towards myeloid output [68]. This so-called emergency myelopoiesis is required to support the peripheral consumption of leukocytes during infection. However, several models of chronic infection have shown that prolonged exposure to inflammatory signals, such as IFN-γ, impairs HSC self-renewal, resulting in hematopoietic dysfunction and pancytopenia [65,66,67]. Likewise, the chronic inflammation that accompanies GvHD may be detrimental to hematopoietic function.

Lastly, excessive cytokine levels in the niche can alter niche function. In response to IFN-γ, mesenchymal stem cells (MSCs), which normally support HSPC quiescence and self-renewal, produce IL-6 to promote myeloid differentiation of HSPCs [55]. Similarly, IFN-γ may be responsible for the increased permeability of bone marrow endothelial cells during GvHD, mentioned above [34,69]. Furthermore, inflammatory cytokines alter the expression of adhesion molecules on stromal cells, which is likely to affect the retention of HSPCs within the bone marrow niche [53]. A better understanding of the complex interactions between the niche and HSPCs, both during normal hematopoiesis and during inflammation, may shed light on how the overexpression of cytokines during GvHD affects both HSPC and niche function.

## 4. Clinical Factors Associated with GvHD Impair Hematopoietic Function

During GvHD, patients are subjected to multiple clinical factors that may pose additional challenges to the hematopoietic system. These include, but are not limited to, immune suppressive therapy, infections and their treatment, and changes in the microbiome (Figure 3).

### 4.1. Immune Suppressive Therapy May Aggravate Hematopoietic Dysfunction

Treatment of GvHD consists of strong immunosuppressive therapy. These drugs are aimed at interrupting the inflammatory cascade to prevent further tissue damage. Initially, GvHD is treated with either local or systemic steroid therapy, depending on severity of disease and organ involvement [7]. Unfortunately, 35–50% of GvHD cases eventually become steroid refractory [70]. These patients require additional therapy, predominantly aimed at inhibiting specific pro-inflammatory cytokines or pathways [57]. However, some of the pathways that are targeted by these drugs may also be vital for hematopoietic function. Therefore, the interruption of individual signaling pathways to prevent GvHD-mediated tissue damage may exacerbate GvHD-induced hematopoietic dysfunction.

#### 4.1.1. Corticosteroids

Corticosteroids form the cornerstone of GvHD therapy. Yet, surprisingly, their effect on hematopoietic function remains largely unknown. Corticosteroids inhibit the expression of pro-inflammatory genes on both innate and adaptive immune cells, resulting in widespread immune suppression [71]. Recently, corticosteroids were shown to increase the expression of apoptotic markers on human HSPCs ex vivo [72]. This effect could be largely reversed by the addition of HSPC-supporting cytokines, such as stem cell factor (SCF) or thrombopoietin (TPO), which are normally supplied by the hematopoietic niche. Notably, corticosteroids may also impair niche function. For example, steroids are known to inhibit the differentiation of osteoblasts, which are key niche components [73]. Altogether, these studies imply that corticosteroids may cause HSPC damage, especially when HSPCs are surrounded by a damaged niche.

#### 4.1.2. IL-2 Inhibitors

Due to its central role in T cell activation and proliferation, IL-2 presents a sensible target to treat GvHD [74]. Calcineurin inhibitors such as ciclosporin and tacrolimus are known to inhibit IL-2 production and have been used for decades for the prevention and treatment of GvHD [7]. More recently, basiliximab, a monoclonal antibody against the IL-2 receptor, has become a treatment option for steroid-refractory GvHD [5].

Nonetheless, a basal level of IL-2 signaling appears to be vital for HSC function [75,76]. Using genetic knockout mice, complete IL-2 deficiency was shown to result in impaired hematopoiesis, including anemia, thrombocytopenia and lymphocytopenia [75]. This hematopoietic dysfunction was found to be secondary to dysfunction of regulatory T cells (Tregs) [76]. Knockout of IL-2 impairs Treg function, leading to an increase in IFN-γ signaling from effector T cells. Overactivity of the IFN-γ pathway resulted in increased HSC differentiation and reduced self-renewal, impairing long-term hematopoietic function [76]. If these findings also apply to humans, inhibition of IL-2 signaling, via calcineurin inhibitors or basiliximab, may be detrimental to HSPC function.

#### 4.1.3. TNFα Inhibitors

As mentioned before, TNFα also plays an important role in the pathogenesis of GvHD [77]. TNFα inhibitors, including etanercept, a soluble dimeric TNF-a receptor 2, and infliximab, a monoclonal antibody against TNFα, are used in the treatment of steroid-refractory GvHD [78,79].

Similar to IL-2 inhibitors, TNFα inhibitors may negatively affect hematopoiesis. However, the impact of TNFα on hematopoiesis remains controversial. Ex vivo treatment of human HSPCs with TNFα has been shown to both inhibit [80,81] and enhance their proliferation [82]. Knockout of TNF receptor in mice also yielded conflicting results, revealing both positive [83] and negative [84] effects of TNFα on HSC survival. One potential explanation for these conflicting results may be the inability of these studies to distinguish multipotent HSCs from committed progenitor cells. Recently, Yamashita and colleagues demonstrated that the in vivo HSPC response to TNFα may depend on their degree of differentiation [85]. In committed progenitors, increased TNFα signaling results in apoptosis. Conversely, quiescent HSCs are protected from TNFα mediated cell-death due to increased NF-kB signaling [85]. Together, these findings highlight the complexity of TNF-α signaling in hematopoiesis. The effect of TNF-α inhibitors may vary between HSCs and more committed progenitors, which makes it difficult to predict their effect on hematopoiesis.

#### 4.1.4. IFN-γ Inhibitors

IFN-γ plays an instrumental role in the activation and effector function of T cells during GvHD [86]. Therefore, blockade of IFN-γ signaling may be an effective strategy to treat GvHD. In fact, ruxolitinib, a Janus kinase (JAK) inhibitor used in the treatment of GvHD, is thought to exert its function by preventing downstream signaling of different cytokines, including IFN-γ [87].

Interestingly, inhibition of IFN-γ signaling may improve GvHD-related hematopoietic dysfunction. During GvHD, persistent IFN-γ signaling appears to have a negative effect on hematopoietic function [88,89,90]. In mice, increased IFN-γ signaling results in reduced proliferation of HSPCs by inhibiting expression of genes encoding the transcription factors Myc and Cyclin D1 [88]. Similarly, ex vivo IFN-γ treatment of human HSPCs results in decreased self-renewal capacity and impairs their engraftment potential in mice [89]. These effects were shown to be mediated via suppression of STAT-5 signaling, which is crucial for HSC stemness [90]. As described earlier, IFN-γ may also contribute to destruction of niche cells [62] and impair niche function [55,69]. Together, these findings provide reasons to believe that inhibition of IFN-γ signaling during GvHD may improve hematopoietic function. Emapalumab, a monoclonal antibody inhibiting IFN-γ, was recently approved as a treatment for hemophagocytic lymphohistiocytosis [91]. This drug may prove effective in the treatment of GvHD, by inhibiting T cell activation and tissue destruction, and potentially reversing GvHD-mediated hematopoietic dysfunction.

### 4.2. Infections and Viral Reactivations Exacerbate Hematopoietic Dysfunction

Another clinical factor that may affect hematopoietic function during GvHD is the increased incidence of infections. Several elements render the HCT recipient suffering from GvHD extremely susceptible to infections: the immaturity of the immune system, that is still reconstituting after transplantation; the immune dysregulation caused by GvHD; the disrupted barrier function of damaged skin and intestine; and the immune suppressive medication. The impact of infections, and viral infections specifically, on HSC function have been reviewed in great detail elsewhere [92,93]. These reviews discriminate four mechanisms by which infections may influence HSCs: via direct infection of HSCs; via direct recognition of the pathogen by HSCs; via pro-inflammatory cytokines released upon infection; or via immune-mediated damage to the niche. Increased cytokine levels and immune-mediated damage to the niche that occur during infection are similar to the effects of GvHD-mediated inflammation on hematopoiesis, and were discussed previously. Below, we will discuss the two additional mechanisms that may exacerbate GvHD-mediated hematopoietic dysfunction upon infection: direct infection of HSPCs or recognition of the pathogen by HSPCs.

Bacterial, fungal and viral infections, as well as reactivations of latent viruses, are all common after HCT. Although direct infection of HSPCs with bacteria or fungi is thought to be rare [94], HSPCs can be the target of viral infection [95,96]. For example, both cytomegalovirus (CMV) and human herpes virus 6 (HHV-6), two of the most common viruses that reactivate after HCT [97,98], have been shown to directly infect HSPCs [95,96]. In vitro infection of HSPCs with either CMV or HHV-6 has been shown to result in impaired hematopoietic function [95,99]. Indeed, in retrospective studies, CMV reactivation is associated with poor graft function after HCT [14,100]. CMV may also infect bone marrow stromal cells, resulting in reduced expression of SCF, essential for HSPC function [101]. HSPCs cocultured with CMV-infected niche cells show reduced proliferating capacity, highlighting the functional impairment of the infected bone marrow niche [99]. Thus, direct viral infection of hematopoietic progenitor cells or the niche may contribute to hematopoietic dysfunction.

HSPCs are also able to sense and respond to pathogens directly, without being infected [92,93]. During GvHD, damage to the epithelial barrier of skin and intestine may result in increased bacterial translocation and exposure of HSPCs to bacterial products [102]. HSPCs express Toll-like-receptors (TLRs) which can bind pathogen-associated molecular patterns (PAMPs). In vitro and in vivo studies have shown that TLR ligation on quiescent HSCs results in increased cell cycle entry [103,104,105]. This increase in proliferation may be detrimental to hematopoietic function in the long term. In mice, chronic TLR-signaling has been shown to result in impaired self-renewal of HSPCs, which were unable to sustain hematopoiesis in serial transplantation [106]. Summarizing, infections may aggravate hematopoietic dysfunction during GvHD. Besides the increased inflammatory response, direct viral infection of HSPCs and chronic exposure to microbial signals may impair hematopoietic function.

### 4.3. Antibiotic, Antifungal and Antiviral Therapy Can Directly Impair Hematopoiesis

Due to the high risk and high mortality of infectious complications, prolonged use of antiviral, antifungal and broad-spectrum antibiotic therapy is common—and necessary—in patients suffering from GvHD [7]. Unfortunately, these therapies pose yet another challenge to the reconstituting hematopoietic system. Anemia, leukopenia and thrombocytopenia are registered toxicities of certain antiviral and antifungal compounds, although the precise mechanisms are still unknown [107,108]. Hematopoietic abnormalities are also a common side-effect of antibiotic treatment, regardless of the type of antibiotic used [109,110]. For a small number of drugs, this effect may be a direct effect on HSPCs. For example, treatment with trimethoprim-sulfamethoxazole may result in neutropenia, by inhibiting folate metabolism in granulocyte progenitors [111]. However, for many antibiotics, hematopoietic dysfunction may be mediated by disrupting the microbial colonization of the gut, as described below [112].

### 4.4. Disruption of the Microbiome May Be Detrimental for Hematopoietic Function

As recently reviewed by Yan et al., the composition and diversity of the intestinal microbiota is vital for proper hematopoiesis [112]. In short, commensal microbiota provide continuous low-level inflammatory signaling, positively affecting HSC function both directly and indirectly by stimulating cytokine production of non-hematopoietic cells [113]. In the setting of HCT, the presence of specific microbial signals has been shown to be important for HSPC engraftment [114]. Strikingly, the presence of specific microbiota species post-HCT is correlated to reduced GvHD-related mortality and even relapse [115,116]. These findings illustrate the relevance of the microbiome in hematopoietic reconstitution and function after HCT.

During GvHD, microbiome disruptions may be caused by bacterial infections and antibiotic treatment, but also by GvHD itself [117,118,119]. In murine studies, while HCT alone results in modest changes in microbiota composition, the onset of GvHD is associated with drastic loss of microbial diversity and expansion of bacterial species that may negatively influence GvHD [117,118]. A reduction in microbiome diversity is common in all HCT patients, likely due to the widespread use of prophylactic and therapeutic antibiotics, and is exacerbated in the presence of GvHD [119]. A recent meta-analysis revealed an association between reduced microbiome diversity after HCT and increased overall and transplant-related mortality [120]. Notably, mice treated with antibiotics show hematopoietic dysfunction, which was demonstrated to be caused by changes in the intestinal microbiome [121]. While germ-free mice or mice treated with antibiotics showed reduced numbers of HSPCs, resulting in reduced hematopoietic output, direct antibiotic treatment of HSPCs in vitro did not affect proliferation or survival. These findings suggest that changes in the microbiome may be responsible for the observed hematopoietic dysfunction. Although the mechanisms behind this phenomenon are still unknown, protection or stimulation of microbial diversity after HCT may be beneficial for hematopoietic function during GvHD.

## 5. Concluding Remarks and Future Directions

Despite continuing efforts to reduce its incidence and improve its outcome, GvHD remains both a common and serious complication of HCT. GvHD is associated with severe and long-lasting hematopoietic dysfunction, which may contribute to the high incidence of infectious and bleeding complications and ultimately, mortality. Prevention or adequate treatment of GvHD as well as GvHD-related hematopoietic dysfunction is needed to address the clinical need to improve the outcome of these patients.

The relative importance of HSPCs versus niche components in GvHD-mediated hematopoietic dysfunction is still unknown, yet crucial to guide treatment. As summarized in this review, GvHD is associated with damage to multiple niche cell types as well as HSPCs. Nonetheless, how this damage translates into overall hematopoietic dysfunction remains to be explored. Resolving this question is difficult, due to the complexity of the BM niche and the multitude of reciprocal interactions between HSPCs and each of the niche cell types [25]. Future in vivo studies, which eliminate or restore the function of individual niche components, are needed to elucidate the role of specific cell types during GvHD-mediated hematopoietic dysfunction and to identify targets to improve hematopoietic function.

Similarly, distinguishing between reversible and irreversible injury to the bone marrow may help choose the appropriate treatment for hematopoietic dysfunction. For example, GvHD-related endothelial dysfunction may be reversed by statin drugs, which were recently shown to improve ex vivo endothelial cell function of patients with poor graft function [122]. Alternatively, extensive cellular destruction may be irreversible and require cellular replacement, via transfusion of additional stem cells or niche components. Early recognition and treatment of GvHD may limit tissue injury and potentially restore hematopoietic function while damage is still reversible. Of particular therapeutic interest is the transfusion of immune-regulatory cells, such as MSCs or Tregs, which could theoretically restore cellular composition, as well as enhance the immune-suppressive function of the bone marrow niche [12,123]. Of note, although MSCs show efficacy in the treatment of steroid refractory GvHD, their capacity to migrate to and reconstitute the bone marrow niche remains subject of debate [124].

While the majority of studies investigating GvHD-related hematopoietic dysfunction focus on immune-mediated damage [13,34,88], the role of clinical factors that accompany GvHD remains largely understudied. Although immune suppressive therapy is necessary to reduce GvHD-mediated inflammation, many of these compounds may inhibit pathways that are necessary for HSC survival or function. Comparative studies, investigating the immune suppressive as well as hematopoietic toxicity of these compounds should be performed to better guide treatment decisions. Similarly, the choice between different antibiotic, antifungal and antiviral medications may be influenced by direct effects on hematopoiesis, as well as their indirect effects via the microbiome. Notably, in this review, we discussed a selection of clinical factors that are most likely to impact hematopoietic dysfunction. However, GvHD results in a plethora or stressors that remain to be explored. For example, both neural innervation [125] and nutritional status [126] are involved in steady-state hematopoiesis. These pathways may be perturbed by GvHD itself, but also by clinicians via dietary restrictions or analgesics [127,128].

Taken together, while our understanding of hematopoiesis and GvHD is slowly growing, much remains unknown. Resolving the complex interplay between GvHD, its related clinical exposures and the regenerating hematopoietic system may ultimately identify therapeutic targets to prevent or treat this self-destructive process.

## Figures and Tables

**Figure 1 cells-10-02051-f001:**
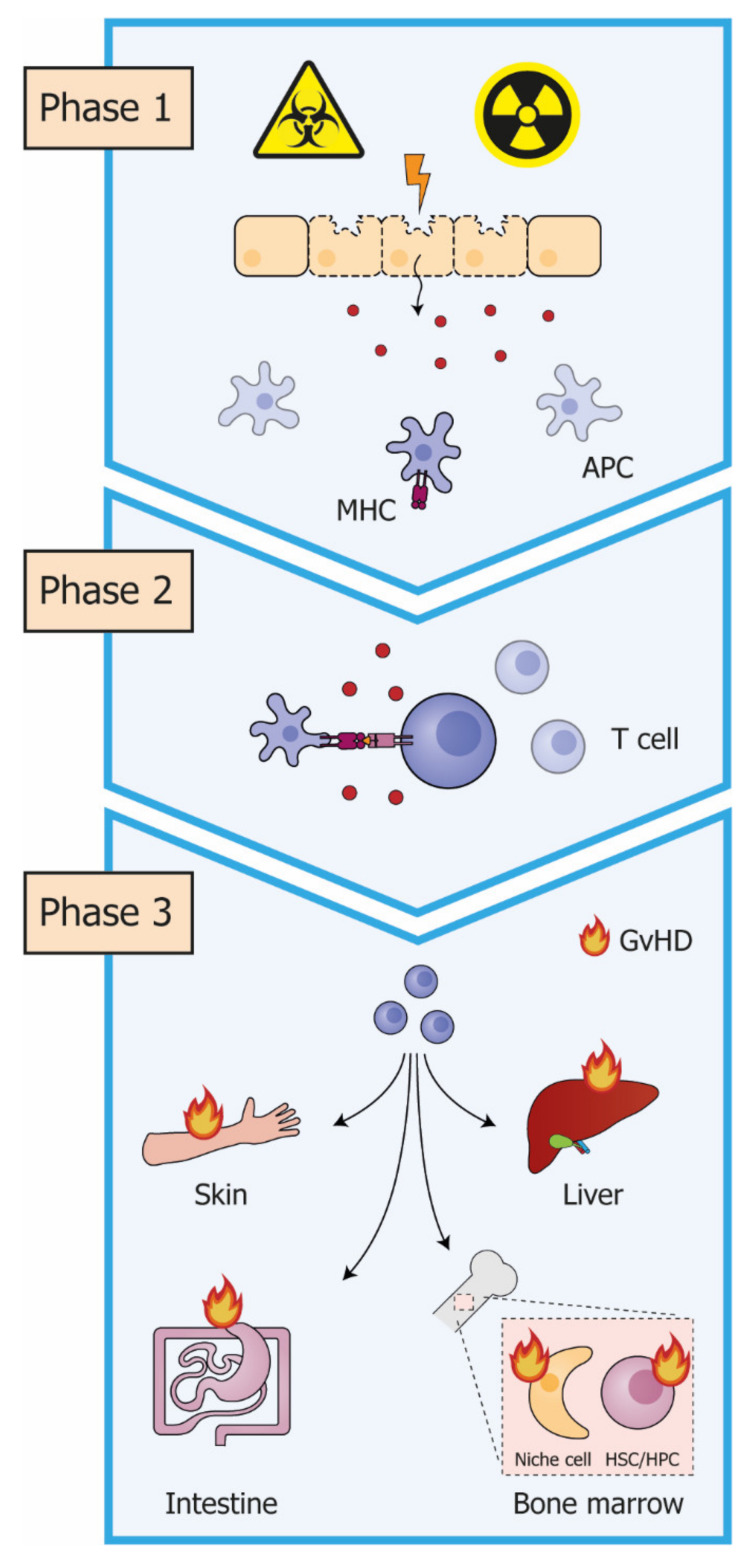
The three phases in GvHD pathogenesis. *Phase 1*: Chemotherapy and/or radiotherapy, given as part of the pre-transplantation conditioning regimen, induce widespread tissue damage. As a result, host antigen presenting cells (APCs) become activated. Activated APCs secrete pro-inflammatory cytokines, increase their expression of major histocompatibility complex (MHC) and costimulatory molecules and migrate to the lymph nodes. *Phase 2:* The combination of inflammatory signals and antigen presentation by APCs initiates an alloreactive T-cell response. *Phase 3:* Activated donor T cells migrate to the target tissues, such as skin, intestine and liver. Here they propagate the inflammatory response and destroy the host target cells. T cells also target the bone marrow, damaging both niche cells and hematopoietic stem and progenitor cells. GvHD: graft-versus-host disease; HSC: hematopoietic stem cell; HPC: hematopoietic progenitor cell.

**Figure 2 cells-10-02051-f002:**
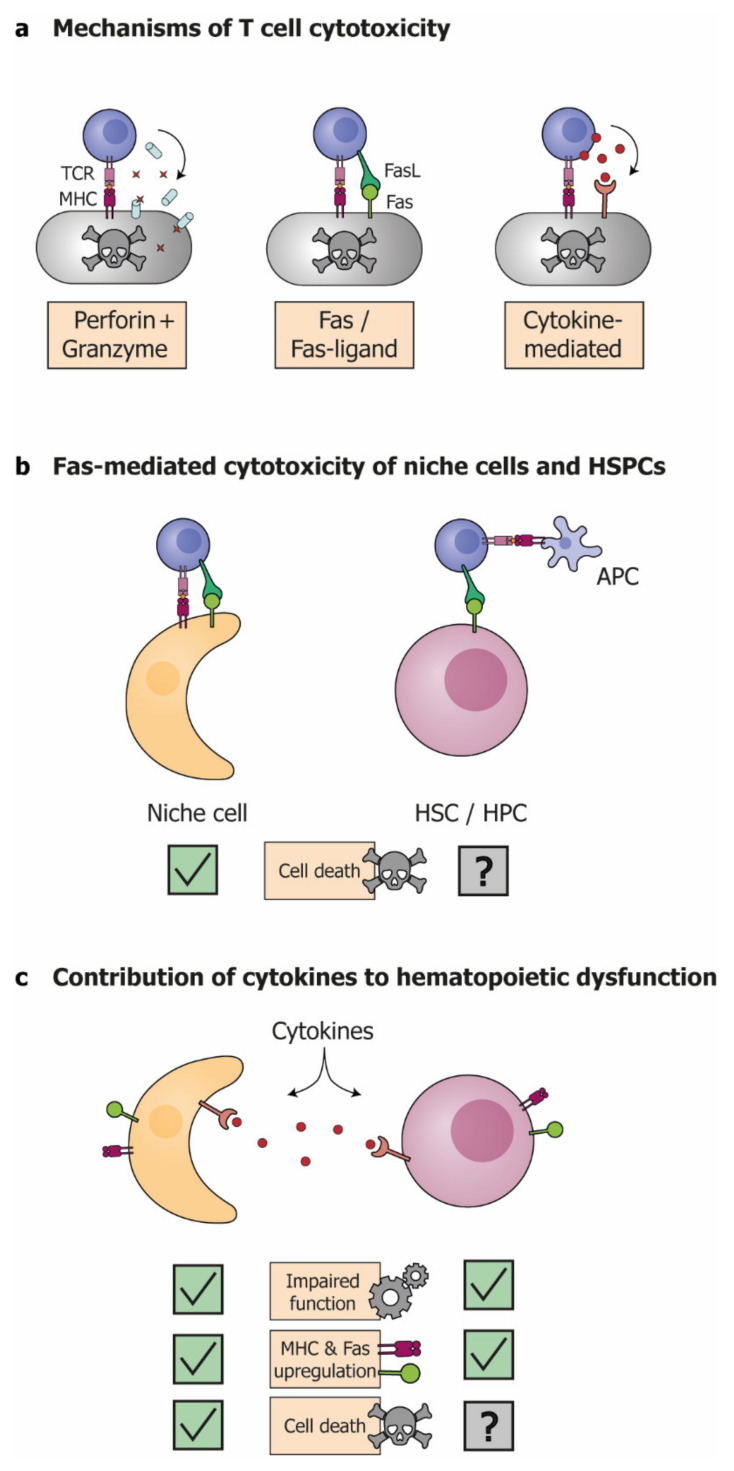
Mechanisms of GvHD-related hematopoietic dysfunction (**a**) T cells can induce target cell destruction via perforin/granzyme, Fas/FasL and cytokine-mediated cytotoxicity; (**b**) Fas-mediated destruction of niche cells contributes to GvHD-related hematopoietic dysfunction. Fas-mediated cytotoxicity does not require MHC expression on target cells, but whether donor HSPCs are destroyed by donor T cells remains subject of debate; (**c**) during GvHD, high cytokine levels may induce apoptosis in niche cells, and potentially in HSPCs. Cytokines increase the expression of MHC and Fas on target cells. Furthermore, cytokines impair the function of niche cells and HSPCs. TCR: T cell receptor; MHC: major histocompatibility complex; FasL: Fas-ligand; APC: antigen presenting cell; HSC: hematopoietic stem cell; HPC: hematopoietic progenitor cell.

**Figure 3 cells-10-02051-f003:**
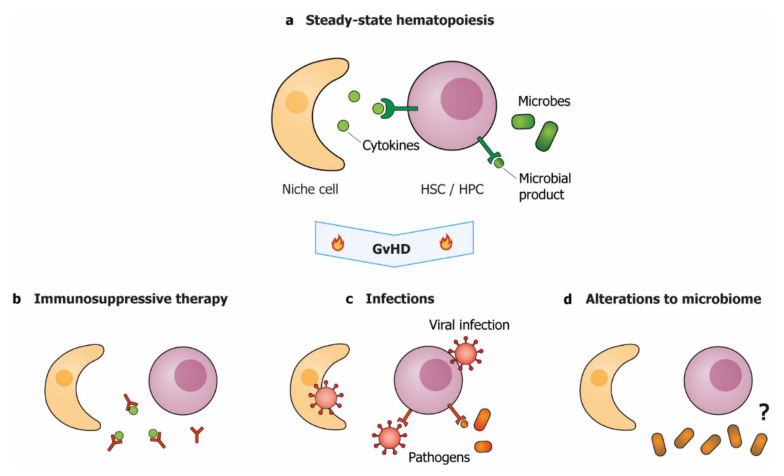
GvHD-associated clinical factors aggravate hematopoietic dysfunction. (**a**) During steady-state hematopoiesis, HSPC function is supported by niche-produced cytokines and low levels of microbial products. During and after GvHD, hematopoietic function may be impaired by; (**b**) immunosuppressive therapies that interfere with cytokine signaling; (**c**) direct infection of niche cells or HSPCs and increased levels of microbial products, or; (**d**) reduced diversity of the microbiome, via a yet unknown mechanism. HSC: hematopoietic stem cell; HPC: hematopoietic progenitor cell.

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
