# Peer review of "Hematopoietic Dysfunction during Graft-Versus-Host Disease: A Self-Destructive Process?"

_cells, 2021, doi:10.3390/cells10082051_

Round 1

Reviewer 1 Report

The authors provide an extensive overview of the processes related to GVHD (and related treatment/complications) affecting bone marrow and hematopoiesis. Although the overview is very complete and the manuscript is well-written, in my opinion it is limited to a summary of relatively well-known facts and theories without any additive value to the currently existing literature as mainly listed in the references by the authors, except for one important review: "Szyska M and Na I-K (2016) Bone Marrow GvHD after Allogeneic Hematopoietic Stem Cell Transplantation. Front. Immunol. 7:118.
doi: 10.3389/fimmu.2016.00118"

Revision is needed to concise and prioritize the current manuscript and to add some novel idea's/suggestions for future research.

Reviewer 2 Report

The manuscript, “Hematopoietic Dysfunction During Graft-Versus-Host Disease: a Self-Destructive Process?” by Muskens, Lindemans and Belderbos reviews foundational and recent literature on hematopoietic failure during GVHD and discuss potential modalities to overcome the deleterious effects and improve hematopoietic function in patients. This is an important and clinically relevant topic, it is therefore important that this scientific topic is reviewed. The review is well organized and written, and covers a lot of ground. Each topic is described accurately, not exceeding in in-depth discussions. The authors cover recent advances in the field as well as controversial topics.

  • While the emphasis is on GVHD-induced hematopoietic dysfunction in the bone marrow, it would be beneficial to mention in the manuscript, perhaps in the introductory paragraph, the detrimental effects of GVHD on thymic function and the consequences on the hematopoietic system. 
  • line 26-27 ("the recipient receives...their own"): correct the grammar 
  • line 41 ("The B cell compartment is the slowest to recover and may take up to several years"): add reference
  • line 312: use italic ("In vivo")
  • paragraph 4.4: please consider including the following references: Jenq, R. R.et al. Intestinal Blautia Is Associated with Reduced Death from Graft-versus-Host Disease. Biol Blood Marrow Transplant 21, 1373-1383, doi:10.1016/j.bbmt.2015.04.016 (2015); Gavriilaki, M. et al The Impact of Antibiotic-Mediated Modification of the Intestinal Microbiome on Outcomes of Allogeneic Hematopoietic Cell Transplantation: Systematic Review and Meta-Analysis. Biol Blood Marrow Transplant, doi:10.1016/j.bbmt.2020.05.011 (2020) 
  • line 456-457: please revise the following sentence to improve clarity "However, there is much to be gained from studies investigating of the interaction between these processes" 
  • Figures are appropriate even though it would be beneficial to better highlight “GVHD” in Fig1 and Fig3.

Round 2

Reviewer 1 Report

I would like to thank the authors for the revisions. As mentioned before, in my opinion this manuscript is a nice overview and well-written, however without highly relevant novel findings/suggestions to the field.